# One Tree at a Time: Restoring Landscape Connectivity through Silvopastoral Systems in Transformed Amazon Landscapes

Karolina Argote [1,2,*], Beatriz Rodríguez-Sánchez [1], Marcela Quintero [1] and Wendy Francesconi [1]

[1] Alliance of Bioversity International and International Center for Tropical Agriculture (CIAT), km 17 Recta Cali-Palmira, Cali 763537, Colombia
[2] Institut Méditerranéen de la Biodiversité et d'Ecologie Marine et Continentale (IMBE), UMR Aix-Marseille Université, CNRS, IRD, Avignon Univ., 13545 Aix-en-Provence, France
[*] Correspondence: karolina.argote-deluque@imbe.fr

**Abstract:** Due to the continued expansion of pastures and illicit crops, the Andes-Amazon foothills in Colombia are one of most threatened biodiversity hotspots in the country. Halting and restoring the connectivity of the landscapes transformed over the last 40 years and now dominated by extensive cattle ranching practices, represents a challenge. Silvopastoral systems (SPSs) have been proposed as a strategy to help conserve the biodiversity by improving landscape connectivity. However, understanding the contributions of SPSs to biodiversity conservation still requires additional research. At the farm scale (here called farmscape), we compared different landscape fragmentation and connectivity metrics under two SPS conditions (with and without). Overall, the adoption of SPSs increased the probability of connectivity (PC) index in all cases. However, the contributions of SPSs to landscape connectivity were not linear. Greater PC increases were observed in highly degraded farmscapes ($\Delta$Pc = 284) compared to farmscapes containing patches that were better connected and had larger habitat areas ($\Delta$Pc = 6). These variables could play a fundamental role in enhancing the landscape connectivity through restoration activities that seek to improve biodiversity conservation. Even if they are relatively small and scattered, in highly degraded cattle ranching systems, SPSs could significantly improve the landscape connectivity, which in turn could improve wildlife conservation.

**Keywords:** Andes-Amazon foothills; fragmentation; connectivity; fragstats; conefor; SPS

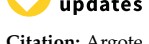



## 1. Introduction

Colombia ranks among the top most biodiverse countries in the world [1]. Yet, the country is facing some of the highest losses in humid primary forests (1.69 Mha lost from 2001 to 2020, corresponding to 36% of its total forest cover loss during the same time period) [2]. As a result, we are witnessing the accelerated habitat destruction of a large number of species across taxonomic groups, from reptiles such as the Amazon river turtle (*Podocnemis expansa*), amphibians such as the golden frog (*Phyllobates terribilis*), birds such as the harpy eagle (*Harpia harpyja*), to mammals such as bats (*Diphylla ecaudata*), the titi monkey (*Callicebus caquetensis*), the jaguar (*Panthera onca*) and the spectacled bear (*Tremarctos ornatus*), all of which considered critically endangered (CR) by the International Union for Conservation of Nature (IUCN) [3]. The Andean-Amazon foothills are a transition area between the Eastern Cordillera of the Andes and the Colombian Amazon. The region provides a clear example of how forest fragmentation and deforestation have greatly reduced the flow of organisms and their genes between ecosystems and populations [4]. According to Colombia's national environmental institute IDEAM, 57% of this transitional area has been converted from natural habitats to agriculture [5]. From 2000 and 2017, the average deforestation rate was approx. 63,000 ha/year, and cattle ranching is the preferred (80%) agricultural activity in the region [5]. The loss of natural habitat puts at risk one of the most important ecological corridors in the continent, which connects four national protected areas: Picachos, Tinigua, Sierra de la Macarena, and Chiribiquete.

In the midst of the Colombian post-conflict era and during the last 5 years (2017–2022), after the national guerrilla group known as FARC (which in Spanish stands for Colombian Revolutionary Armed Forces) left the territory, the situation for Colombia's biodiversity is even more uncertain. The expansion of grasslands for cattle ranching and monoculture agriculture continues to increase at an alarming rate and at the expense of native forests and other species-rich ecosystems [6,7]. The development and implementation of post-conflict economic plans in the region are contributing to increases in deforestation [8]. Many of these plans are based on the exploration and extraction of minerals and hydrocarbons [9,10]. The stable generation of financial resources and income for local communities is also a key objective in the region. While the plans are meant to bring development into what has been an impoverished region, the problem may lay in the poor transition between government administrations, to make these plans sustainable in nature [11]. What's worse, is that new armed groups have emerged and are invading the region. These groups are carrying out illegal activities, such as growing illicit crops, and illegal mining and logging. The complex and multidimensional factors in the Andes-Amazon foothills has and continues to make the implementation of appropriate sustainable development and biodiversity conservation strategies very difficult in the region. Putting forward options for maintaining and improving landscape connectivity is one of the challenges for biodiversity conservation in highly fragmented landscapes [12]. As a main deforestation driver, cattle ranching in the Andes-Amazon foothills is characterized by poor water and soil management, resulting in very low grass productivity (0.62 head ha$^{-1}$) [13]. An attractive alternative would be to work with the local communities to increase their environmental awareness and become land stewards through the adoption of silvopastoral systems (SPSs) [14]. SPSs are multifunctional agroforestry practices that intentionally combine cattle ranching production with grasses, legumes, and trees, to produce fodder and forage, as well as timber and fruits or nuts in some cases [14–16]. Hence, compared to conventional pastures dominated by monocultures, SPSs optimize land productivity and contribute to conserving water, soil, and nutrients. SPSs could integrate their components in a mutually beneficial way, by providing nutrient recycling and by enhancing productivity, animal welfare, soil retention, and carbon sequestration, which results in greater income and well-being for farmers. Thus, these systems benefit producers and society as a whole at the local, regional, and global scale [17,18].

The above-mentioned benefits of SPSs have been well documented in the literature [18,19]. However, understanding the contributions of SPSs to biodiversity conservation still requires additional research [20,21]. It would be valuable to identify the criteria that could optimize the impacts of SPSs in increasing landscape connectivity for biodiversity conservation. We hypothesize that the contributions of SPSs to structural connectivity vary as a function of fixed-variables (i.e., size, location, shape) that could be manipulated during the design of SPSs. In other words, site-specific planning and implementation of SPSs could potentiate the impact of these to increase the landscape connectivity for conservation purposes. The research intends to better understand the potential conservation benefits of livelihood practices that work in synergy with the interests and preferences of farmers.

To address the hypothesis, we conducted a three-level research approach to: (i) understand the deforestation trends in the Colombian Amazon ecoregion during a twenty years period (2001–2021) (1st Level), (ii) estimate the landscape fragmentation changes in the Andean-Amazon foothills of Colombia (2nd Level), and (iii) compare the landscape connectivity and the fragmentation metrics under two SPS conditions (with and without) at the farm scale (3rd Level).

## 2. Materials and Methods

### 2.1. Study Area

The study was carried out in one of the most deforested areas of the Colombian Amazon (1st level), located at the transition between the Eastern Cordillera of the Andes and the Amazon. This area is characterized for its biodiversity and high levels of endemism,

as well as by its cultural significance. At the same time, it is one of the most threatened areas due to habitat degradation, loss of biodiversity, disruptions to the water cycle, social pressure on natural resources, and forest fires [22]. The deforestation frontier in this region has been rapidly advancing from the Andes Mountain to the deep Amazon. By 2020, deforestation in this ecoregion has reached 11,519 km$^2$ [23].

Within this area, we selected a 41 × 54 km window (2214 km$^2$) located in the Andean-Amazon foothills to the northwest region of the department of Caquetá, between 1°05′ N, 76°02′ W and 1°27′ N, 75°33′ W (2nd level). Finally, we selected 12 analysis sub-windows of 3 × 3 km (9 km$^2$) containing 24 farms where SPSs had been implemented (Figure 1).

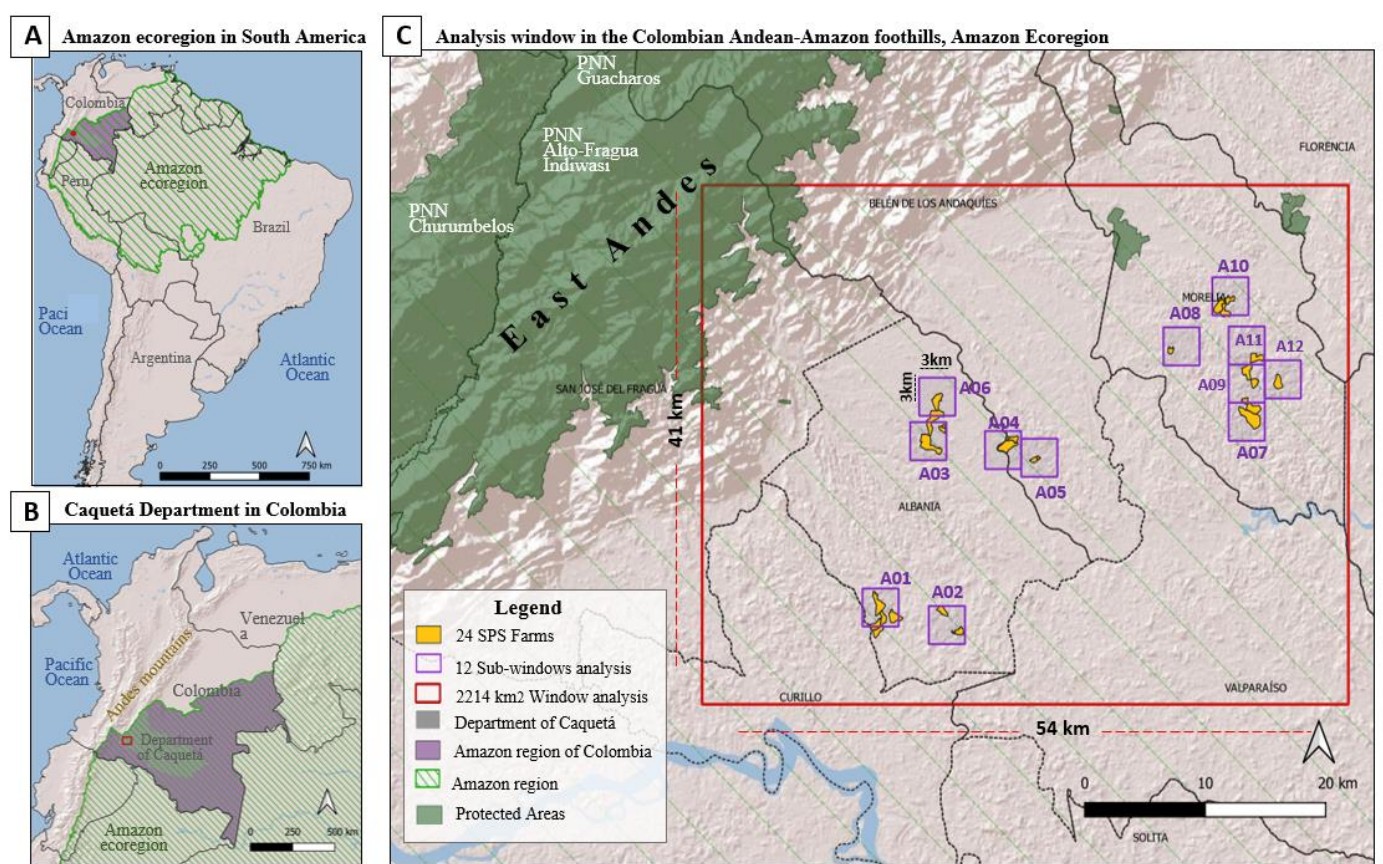

**Figure 1.** Study Area. (**A**) Location of Amazon ecoregion in South America; (**B**) Location of Caquetá Department in Colombia; (**C**) Location of the analysis sub-windows and the 24 SPS implementation farms.

The selected study area is a predominantly agricultural landscape connected through a network of roads and small urban areas. Within this target region, habitat restoration was proposed through the implementation of the Sustainable Amazon Landscapes (SAL) project, an initiative that engaged and co-designed the restoration of cattle ranching pastures, in partnership with farmers and local environment organizations. The project promoted wildlife conservation through the adoption of silvopastoral systems in combination with the conservation of valuable natural areas such as forests, secondary vegetation, and water bodies within farms. In 2018, SPSs were implemented in a total of 24 farms. Each SPS system was designed according to the type of farm, their natural areas, their water accessibility, their land uses, and production activities, among other variables. As part of the SAL project, not only living fences and scattered paddock trees were established as SPSs, but practices such as: paddock rotation, forage banks, protection of secondary forest areas, and bodies of water to promote natural regeneration, were promoted along with other sustainable practices co-designed between experts and farmers.

Two types of silvopastoral systems were established on these 24 farms. One system was the so-called intensive system composed of plots of *Brachiaria decumbens* in association with Kudzu (*Pueraria phaseoloides*) mainly, interspersed with strips of timber trees inside and around the edge of the plot, mainly shade trees, such as Chilco (*Miconia elata*), Madura-plàtano (*Jacaranda copaia)*, Nogal (*Cordia alliodora*), and shrub species such as the Boton de oro (*Tithonia diversifolia*). The second system was a simpler SPS, composed of a strip of timber trees, such as Fono (*Eschweilera andina*), Lacre (*Vismia baccifera*), Melina (*Gmelina arborea*) planted at the edge of the plot (living fence) and interspersed strips of *Tithonia diversifolia* and *Brachiaria decumbens*. In both cases, the lines of trees were planted at a distance of between 6 m and 8 m (depending on the species or combination of species used) [24]. Native tree seedlings produced by the nursery of the Centro de Investigaciones Amazónicas Macagual at the Universidad de la Amazonia (project implementing partner) were used for planting. The survival rate of the seedlings depended on the species chosen in the design of the silvopastoral system on each farm, but on average, this value ranged between 70 and 80%. Additionally, larger plants (height between 60 cm and 90 cm) were transplanted to increase their probability of survival in the field. The silvopastoral systems on all farms were planted during the same period (between October and December 2018).

*2.2. Methodology*

Our methodology is made up of three types of analysis at three different scales, going from broad to focus (Figure 2):

1.  The first level analysis provides the contextual information about deforestation trends between 2001 and 2021, using the annual maps of deforestation areas for the entire Colombian Amazon ecoregion.
2.  The second level analysis corresponds to a temporal analysis of the landscape fragmentation, using the land use/land cover maps (LULC) for 2002 and 2018, generated by the Amazonian Institute of Scientific Research (SINCHI) [25,26]. This analysis was carried out on a 2214 km$^2$ window located in the Colombian Andean-Amazon foothills.
3.  The third level analysis compares the probability of the landscape connectivity in 2018, under two SPS conditions (with and without the implementation of silvopastoral systems). This analysis was carried out at the farm level (here called farmscape) on 12 sub-windows each of 3 × 3 km.

2.2.1. First Level Analysis: Deforestation Trends in the Amazon Ecoregion (2001–2021)

The Colombian Amazon ecoregion covers about 6.8% of the entire Amazon rainforest biome in South America. In Colombia, the ecoregion is located in the southeast of the country, representing 42.3% of the national territory. Yet, only 12% is protected under the country's National Park System, and about 46% of this percentage is designated as indigenous territories [27]. In Colombia, the Amazon ecoregion covers 10 of the 32 departments in the country, including Amazonas, Guaviare, Caquetá, Vaupés, Guainía, and to a lesser extent, Vichada, Meta, Putumayo, Nariño, and Cauca. To gain insights into the deforestation trends in the ecoregion, we used annual deforestation maps produced by the GLAD laboratory [2].

Among the GLAD products, annual forest cover loss maps (here called forest cover loss) are generated using a supervised learning algorithm that processes Landsat (TM/ETM+) satellite images at a 30 m spatial resolution. The classifications were implemented at per-Landsat pixel level, with a minimum mapping unit equivalent to 0.09 ha. The forest loss is mapped as a single dynamic class using a supervised bagged classification tree algorithm. The training data served as the dependent variable and the 1985–2000-time interval metrics as the independent variable in the tree model. The lab applies the classification trees yielding a map depicting the forest loss between 2001 and 2021. GLAD defines forest loss as the disturbance or complete removal of the tree cover canopy (below 25% tree canopy cover) [2]. This means that any conversion of natural forests, be it plantations, selective logging, or shifting the cultivation practiced by local communities, would be considered forest loss. The

model accuracy represented by the commission error (false alerts), ranged from 95.5% ± 1.8 to 97.2% ± 1.7 in primary and secondary forests, and the omission errors (missed alerts), ranged from 82.6% ± 21.5 to 57.5% ± 8.3 in primary and secondary forests [28,29].

These data were downloaded for each year between 2001 and 2021, and using QGIS [30], were cut for the Amazon ecoregion. Areas in the raster reported annually as new alerts of primary or secondary forest cover loss, were calculated. Additionally, and in order to improve the accuracy of the model by reducing the number of false deforestation alerts, the official national forest/non-forest 2001 layer for Colombia, generated by IDEAM, was used as a mask. All alerts reported outside of this area were removed.

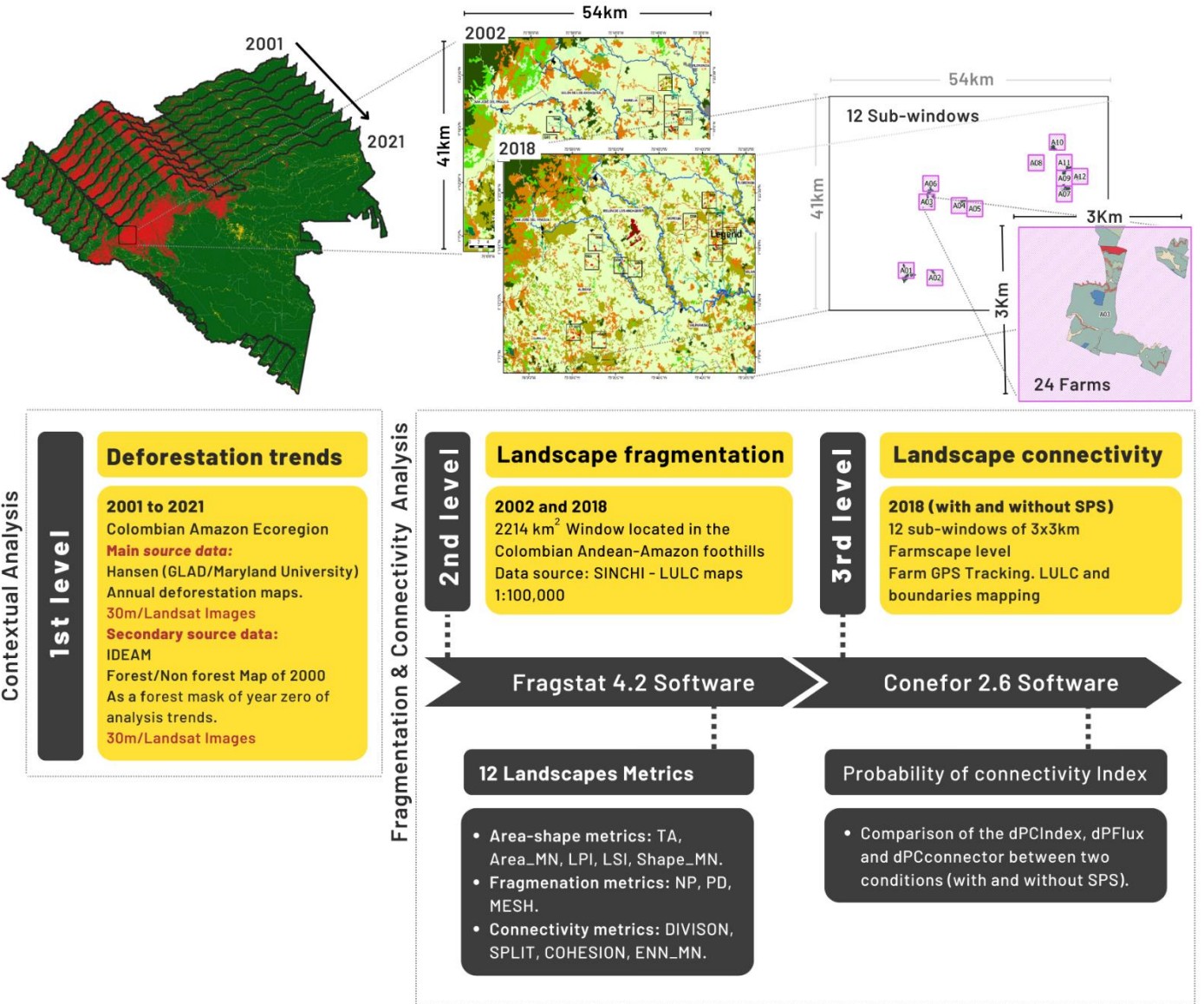

**Figure 2.** Flowchart of the research methodology. It shows the three levels of analysis addressed in the methodological approach: (1) Deforestation trends in the Colombian Amazon ecoregion; (2) Landscape fragmentation in a 2214 km² window in the Andean-Amazonian foothills; and (3) Comparison of landscape connectivity with and without SPS implementations.

2.2.2. Second Level Analysis: Landscape Fragmentation

Definition of Landscape Units

To reduce the heterogeneity within the 2214 km² analysis window, the area was subdivided into landscape units. These units shared similar characteristics in: relief, geo-

pedological features, climate, biome, and physiography [31]. This was accomplished using the QGIS software by merging all of these geographic layers into a single layer. The input data were downloaded in vector format scale 1:100,000 from the WFS (Web Feature Service) of the Agustin Codazzi Geographic Institute of Colombia (IGAC) [32].

Finally, using the resulting landscape units' polygons, the 2002 and 2018 land cover maps were cut using QGIS in order to create the inputs layers for FRAGSTATS. The fragmentation analysis in FRAGSTATS was run by landscape.

Fragmentation Metrics

At the landscape level, a temporal analysis was conducted in the 2214 km$^2$ window for the four landscape units identified above. We used the software FRAGSTATS Version 4 [33], to quantify and evaluate the changes in the structural attributes, comparing fragmentation in 2002 vs. 2018. The fragmentation analysis was based on 12 metrics, but only those with the largest changes were reported in the results. The 12 metrics were grouped into the following three different categories, based on the type of information assessed:

- **Area-shape metrics:** total area (TA), mean patch area (Area_MN), largest patch index (LPI), landscape shape index (LSI), mean shape index (shape_MN).
- **Fragmentation metrics:** number of patches (NP), patch density (PD), effective mesh size (MESH).
- **Connectivity metrics:** landscape division index (DIVISION), splitting index (SPLIT), patch cohesion index (COHESION) and Euclidean Nearest Neighbor Distance (ENN_MN).

Generally, forest-non-forest layers are used as input for landscape fragmentation analyses [33]. However, not in this case since we are talking about highly fragmented areas, with a very low percentage of forest and where the dominant land use is pastures. Thus, in these degraded landscapes, not only primary forests could have a high ecological value, but also secondary forests, gallery and riparian forests, secondary vegetation, and established silvopastoral and agroforestry systems. Under this assumption, for the fragmentation and connectivity analyses, the LULC maps by the SINCHI Amazon Research Institute at a 1:100,000 scale [25,26] were used to prioritize not only the forest areas but also the areas that could be used as habitat for wildlife. The available LULC maps of the Colombian Amazon, for the years 2002 and 2018, were also used to compare the landscape's fragmentation changes over the 15-year period. Land covers and land uses that could be used as habitat by native species (defined as "crucial areas") were used as foreground, and other land-use types, such as pastures and crops, were assigned as the background. In order to rank the coverages, the concept of naturalness defined by Machado was used. Machado's (2004) naturalness index (NI), was used to define the level of naturalness of the different land cover classes [34]. The NI uses a 0–10 ranking system to define a minimum to a maximum relative degree of naturalness for a particular site (Table 1). In the present study and based on Machado's ranking, land cover/land use patches with a NI equal or greater than 4 (NI ≥ 4), were assumed to provide habitat to native species, hence contributing to their conservation within the landscapes.

**Table 1.** Naturalness index (NI) ranking by land cover/land use. The "crucial areas" used in the fragmentation and connectivity analysis correspond to land covers with a NI value equal or greater than 4 (NI ≥ 4). Photos by Argote, K.

| Naturalness Categories | Land Cover/Land Use | Rank | Land Use Validation Photographic Evidence |
|---|---|---|---|
| Natural system; dominance of wild native species, few exotic invader species; minimal artificial infrastructure, temporary or removable. | Dense Highland Forest Dense Highland Floodplain Forest Natural Water Bodies Natural Sandy Areas Natural Marshy Areas | 9 |  |
| Sub-natural system; presence of wild native species and possible extended presence of exotic invader species, but not dominant (low impact); artificial elements restricted, not widespread. | Fragmented Forest with Pastures and Crops Fragmented Forest with Secondary Vegetation | 8 |  |
| Quasi-natural system; extensive anthropic activities of low physical impact; presence of wild native species but also exotic invader species well established but not dominant; natural structures modified but not distorted. Little alteration of water dynamics | Riparian Forest | 7 |  |
| Semi-natural system; presence of wild native species and possible extended presence of exotic invader species. Occasional addition of energy and/or extraction of renewable resources or of non-relevant materials. It may include abandoned cultural systems undergoing natural recovery. | Secondary Vegetation | 6 |  |
| Cultural assisted system that combine tree growing, forages, and shrubs/trees with the production of livestock. Presence of wild native species and possible extended presence of exotic invader species. Natural elements intermixed with artificial ones, as managed wooded pastures, in patches or corridors. Active management of the water cycle. | Silvopastoral Systems | 4 |  |
| Highly intervened system: permanent areas with agricultural production. Natural biodiversity is severely reduced; its elements are isolated (intense fragmentation). | Mosaic of crops, pastures, and natural Spaces Pasture Mosaic with Natural Spaces Pastures and Crops Mosaic | 3 |  |
| Transformed system; anthropic processes governing; clear dominance of artificial elements; frequent intensive vertical development; vestiges of natural elements. | Oil Palm Weedy Pastures Open Treeless Pastures | 1 |  |
| Artificial system | Hydrocarbon Exploitation Urban Areas Bare and Degraded Lands | 0 |  |

2.2.3. Third Level: Landscape Connectivity at the Farmscape Level

The third level analyses were carried out at the farmscape level. A total of 12 sub-windows (3 × 3 km) were created within the prioritized landscapes (Figure 1C). The sub-windows represented real farm and surrounding landscape conditions to carry out the landscape connectivity analysis simulating the implementation of SPSs in 2018. The location of the sub-windows was defined based on the location of the 24 farms where the SPS interventions were carried out through the SAL project initiative. For each of the 24 farms participating in the initiative, the plot areas for the SPSs were planned and mapped in a participatory manner. Basically, the farmers decided where and how to implement the SPS interventions in their farms. The location and extent of the SPS polygons were measured on site with a Geographic Positioning Systems device. Three types of SPSs were adopted by farmers: (i) living fences, (ii) pastures with scattered trees, and (iii) densely planted patches (<0.5 ha) with shrubs and trees with edible foliage to feed cattle, known as forage banks. Among these, only living fence areas and pastures with scattered trees were considered for analysis. Forage banks were not considered due to their small patch size (<0.5 ha).

Using Machado's land cover classification, the SPS areas were considered "managed wooded pastures" and ranked with a naturalness index of four (NI = 4). As land cover areas with NI $\geq$ 4 were considered important in providing habitat to wildlife, the SPS areas were added to the 2018 "crucial habitat" layer developed in the previous landscape fragmentation analysis (2nd Level). The probability of connectivity (PC) index was estimated using CONEFOR [35] for the "crucial areas" layer with and without the SPS polygons in each of the sub-windows. The PC is defined as the probability for two animals located at random in the landscape, to be found in habitat areas that are connected. The PC (1) is measured using the following equation:

$$PC = \frac{\sum_{i=1}^{n} \sum_{j=1}^{n} a_i a_j p_{ij}}{A_L^2} \qquad (1)$$

where, the *PC* index is the sum of the probability of interconnections ($p_{ij}$) among the number (*n*) of habitat patches "*i*" through "*j*", and the areas of those patches "$a_i$" and "$a_j$", over the total landscape area ($A_L$). The total landscape area comprises both habitat and non-habitat patches [36].

Following the method described by Saura and Rubio in 2010 [35], the value of each patch in increasing connectivity (i.e., the dPC value) was divided into three components to distinguish habitat availability and connectivity as follows:

$$dPC = dPCintra + dPCflux + dPCconnector, \qquad (2)$$

As illustrated in Equation (2), dPCintra represents the surface area of a patch, dPCflux represents the area-weighted dispersal flux between patches, and dPCConnector represents the contribution of a patch as a connecting element or a stepping stone which allows to maintain the connectivity between patches [37].

The impact of SPSs on landscape connectivity will be specific to the habitat conditions found in each sub-window. To mitigate the variability caused by the unique distribution and the availability of "crucial areas", and to simplify the analysis of the potential contributions of SPSs under different farmscape conditions, the 12 sub-windows were grouped based on the results from the PC index and landscape fragmentation metrics. Six variables were used to conduct a K-Means cluster analysis. All analyses were performed using R Statistical Software (v4.1.2) [38]: (i) change in PC (ΔPC): difference in dPC index in each sub-window between the two conditions, with and without SPSs; (ii) change in the number of patches (ΔNP): difference in the NP by sub-window between the two conditions, with and without SPSs; (iii) change in the total habitat area by the sub-window (ΔCA): difference in the valuable area in the sub-windows (crucial areas) between the two conditions); (iv) size of the largest patch (ΔLP) in 2018 with SPSs by the sub-window; (v) change in the average

of patch crucial-areas; (vi) Naturalness value by the sub-window in 2018. A silhouette coefficient was calculated to determine the optimal number of clusters. Finally, a one-way analysis of variance (ANOVA) was conducted using the sub-window clusters to compare the impacts of SPSs under different landscape conditions.

## 3. Results

### 3.1. First Level Analysis: Deforestation Trends in the Amazon Ecoregion (2001–2021)

As seen in Figure 3, the deforestation in the Colombian Amazon tripled in the last two decades. In 2001, the annual rate of forest cover loss in the Colombian Amazon was 74,000 ha per year. Despite the fact that a historical low was reached in 2003 with 34,000 ha/year, by 2007 deforestation doubled the 2001 values and quadrupled those in 2003 with 129,000 ha/year. By 2008, forest cover loss decreased and stabilized at an average annual rate lower than 60,000 ha/year. In 2018, Colombia reached a historical high of 250,000 ha of forest cover loss in the Amazonian ecoregion. More recent deforestation values continue to be high, surpassing 180,000 ha/year (data for 2021 is only for the first half of the year).

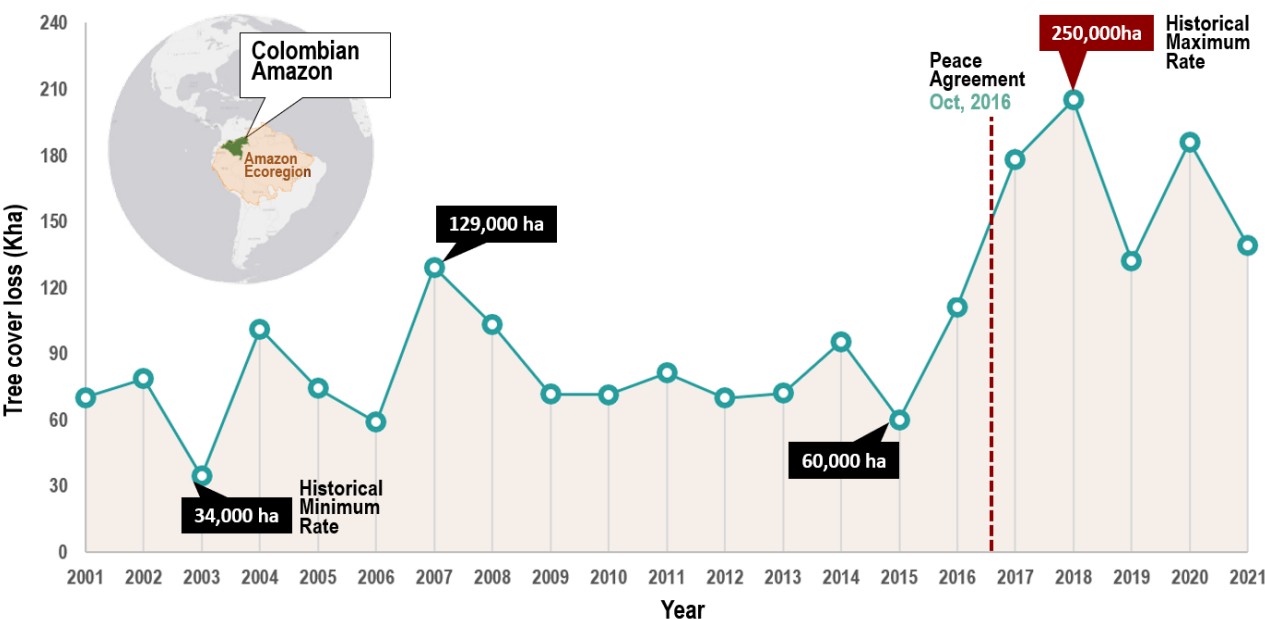

**Figure 3.** Colombian Amazon deforestation between 2001 and 2021, based on Hansen-University of Maryland geospatial raw data [2].

### 3.2. Second Level Analysis: Landscape Fragmentation

<u>Definition of Landscape Units</u>

As seen on the map (Figure 4A), four landscapes units were identified:

i.   **Mountain Landscape:** corresponds to the highest and most mountainous parts of the Amazon at the foot of the Andean region with slopes of 25–50%. Its forests have ecological and ecosystem characteristics of great importance, presenting high levels of diversity of fauna and flora species. This landscape is dominated by dense highland rainforest, fragmented woods, secondary vegetation, and small agricultural areas.

ii.  **Mountain foot Landscape:** represents a transition between the Andes and the Amazon plain, with slightly wavy reliefs and slopes >25%. Loamy-sandy and loamy-clay soils with good drainage and low fertility. Entisols, Inceptisols, and Ultisols dominate. This landscape is dominated by clean pastures and recent oil palm monocultures.

iii. **Lowland Landscape (lomerio Landscape):** have plain to wavy reliefs with slopes of 3–7% and 12–25%, drainage ranges from imperfect to excessive. Soils formed by sedimentary rocks of the tertiary, moderately deep, well drained, clayey texture, low

fertility, very acidic, low saturation base, low content of organic matter. Dominance of Oxisols and Ultisols. This landscape is dominated by pastures, mosaics of pastures with agriculture and secondary vegetation. The main economic activity is extensive livestock raising.

iv.	**Floodplain Landscape:** are part of the floodplain of rivers that are born in the Andes mountain range. Flat relief, with slopes of 0–3%; they suffer occasional floods every 3 or 7 years. Clay soils with a dominance of Entisols and Inceptisols with poor drainage; superficial and limited by the water-table; fertility is average. This landscape is dominated by pastures and crop-fields.

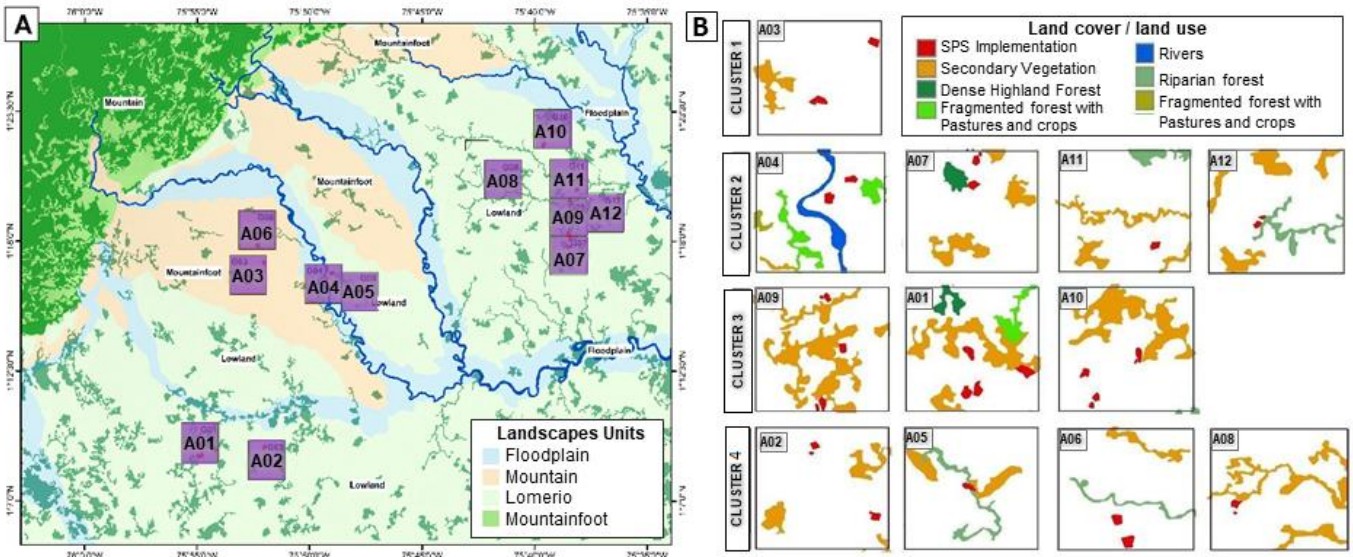

**Figure 4.** Results of clustering with fragmentation the six-landscape connectivity and connectivity metrics/indexes: (**A**) 2214 km$^2$ window analysis with the landscape types and twelve 3 × 3 km sub-windows; (**B**) land cover/land use grouped by cluster.

Following the identification of the landscape units, the landcover maps from 2002 and 2018 were cut for each landscape polygon area. A detailed description on the landscape units and land cover/land use areas can be found in Supplementary Materials (Tables S1 and S2).

Fragmentation metrics

Here we describe the most relevant landscape fragmentation results from the FRAGSTATS temporal analysis, comparing the changes in the structural attributes, between two periods (2002 vs. 2018) in the four defined landscape units. Yet, all results by land-cover and by landscape can be found in Supplemental Materials (Tables S3 and S4):

- **Area-shape metrics:** In the lomerio landscape, the number of patches classified as "secondary vegetation" in 2002, increased by 31% compared to 2018. In contrast, the number of patches classified as "dense highland forests", decreased by 58%. In the mountain landscape, 71% of all forest types ("dense highland forests", "fragmented forest", and "riparian forest") reported in 2002, were lost in 2018. With a lower percentage, in the foothill landscape, 35% of all forest types were lost.

- **Fragmentation metrics:** The mountain landscape was the most fragmented area during the 15 years of analysis. In this landscape, the Effective MESH Size metric decreased by 48% between 2002 and 2018 in the landcover classified as "dense highland forest", and by 99% in the landcover type called "fragmented forest". Similarly, in the lomerio landscape, the Effective MESH Size metric decreased by 75% in the landcover classified as "dense highland forest". This indicates that the primary forests ("dense highland forest") were considerably reduced between 2002 and 2018, both in the mountain

landscape and in the lomerio landscape. Two landscapes with completely different dynamics and drivers of change.

- **Connectivity metrics:** The Euclidean Nearest Neighbor Distance (ENN_MM) metric increased in several forest covers in all landscapes. For example, in floodplain landscape, this metric increased in the riparian strips by 40% and in the fragmented forest, by 43%. In the same way, in the lomerio landscape, the same metric, increased by 34% in the "dense floodplain forest". In the mountain landscape, the ENN_MM metric increased by 46% in the "dense highland forest" and by 67% in the "fragmented forest". Last, in mountain foothill landscape, the ENN_MM metric increased by 16% in the dense highland forest and by 91% in the fragmented forest.

The results show that across landscapes, the degree of isolation (distance between landcover patches within each landscape) has increased in all forest areas. However, the two most fragmented landscapes during the 15-year analysis were the lomerio and the mountain landscape.

### 3.3. Connectivity Comparison with and without the SPS Adoption at the Farmscape Level

3.3.1. Probability of Connectivity Index (PC)

The results for the CONEFOR-estimated PC index are available in Table 2. The table shows the sum of the PC values for all of the habitat patches in each of the 12 sub-windows (farmscapes).

**Table 2.** Results of the dPC index in the two analyzed conditions (with and without SPSs) by subwindows of 3 × 3 km grouped by cluster. dPC corresponds to the sum of dPCIntra, dPCflux, and dPCconnector. The other variables in the table correspond to the descriptive variables of the subwindows such as: CA (the crucial total area in the sub-window), NP (total number of patches in the sub-window, and Shape area (average patch area in the sub-window).

| Cluster | Sub-Window | Without SPS (Before Implementations) | | | | | | | With SPS (After Implementation) | | | | | | | |
|---|---|---|---|---|---|---|---|---|---|---|---|---|---|---|---|---|
| | | CA | NP | Average_PA | dPCintra | dPCflux | dPCconnector | dPC | Area_SPS | CA | NP | Average_PA | dPCintra | dPCflux | dPCconnector | dPC |
| 1 | A03 | 49.75 | 1 | 49.75 | | | | | 9.3 | 59.1 | 3 | 19.7 | 80.8 | 38.5 | 0.0 | 119.3 |
| 2 | A04 | 150.78 | 6 | 23.83 | 33.1 | 133.7 | 6.0 | 172.9 | 7.8 | 158.6 | 8 | 18.9 | 29.2 | 141.5 | 9.7 | 180.5 |
| | A07 | 124.89 | 4 | 29.58 | 42.7 | 114.6 | 0.1 | 157.4 | 6.6 | 131.5 | 7 | 17.8 | 37.8 | 124.5 | 2.5 | 164.8 |
| | A11 | 116.57 | 5 | 22.73 | 35.9 | 128.3 | 3.4 | 167.6 | 2.9 | 119.4 | 6 | 19.4 | 34.3 | 131.3 | 3.5 | 169.1 |
| | A12 | 153.46 | 5 | 30.23 | 28.4 | 143.2 | 4.1 | 175.7 | 2.5 | 155.9 | 7 | 21.9 | 27.4 | 145.1 | 4.1 | 176.7 |
| 3 | A01 | 230.08 | 6 | 35.80 | 46.3 | 107.4 | 2.6 | 156.3 | 19.8 | 249.9 | 10 | 23.0 | 38.5 | 123.0 | 4.2 | 165.7 |
| | A09 | 225.71 | 4 | 54.19 | 63.0 | 74.1 | 0.2 | 137.3 | 8.9 | 234.6 | 9 | 25.1 | 58.1 | 83.7 | 1.1 | 143.0 |
| | A10 | 183.07 | 4 | 43.57 | 99.5 | 1.1 | 0.0 | 100.5 | 8.8 | 191.9 | 9 | 20.3 | 91.9 | 16.2 | 0.2 | 108.2 |
| 4 | A02 | 128.92 | 4 | 16.08 | 63.3 | 73.3 | 0.0 | 136.7 | 5.5 | 134.5 | 7 | 10.0 | 55.0 | 90.0 | 0.7 | 146.0 |
| | A05 | 105.50 | 4 | 17.20 | 33.0 | 134.0 | 4.9 | 171.9 | 2.1 | 107.6 | 5 | 2.1 | 22.9 | 154.2 | 7.1 | 184.2 |
| | A06 | 65.80 | 2 | 28.00 | 61.0 | 77.9 | 0.0 | 139.0 | 9.7 | 75.5 | 4 | 16.4 | 47.7 | 104.6 | 2.6 | 154.9 |
| | A08 | 136.03 | 5 | 26.70 | 48.3 | 103.3 | 1.9 | 153.6 | 2.7 | 138.7 | 7 | 19.4 | 46.4 | 107.2 | 2.4 | 156.0 |

As can be seen in the table, in all of the sub-windows, the dPC values increase when including the SPS implementations (with SPSs). This was expected, as the probability of connectivity in the landscape will increase if new habitat patches are introduced. However, according to the results, the specific contributions to the landscape connectivity by SPSs will vary depending on: the number of patches that are introduced, the arrangement of these patches within the landscape, and the habitat amount.

3.3.2. Landscapes Metrics and PC Index

Using the selected landscape connectivity (PC Index) and the relevant fragmentation metrics from the previous assessments, the k-means cluster analysis and the silhouette score, identified four groups (Figure S1A). The total variance in the data set was 78.2%, and the within cluster sum squares by cluster, was 0, 6.01, 2.65, and 5.71, for clusters one to four, respectively. Yet, according to the one-way ANOVA analysis, each cluster was significantly different to the others in terms of the six landscape connectivity and fragmentation metrics compared: (1) change in the natural area [$F_{(2, 8)} = 110.1$, $p = 0.000001$]; (2) change in the number of patches [$F_{(2, 8)} = 4.584$, $p = 0.0471$]; (3) change in the probability of connectivity [$F_{(2, 8)} = 9.113$, $p = 0.00866$]; (4) biggest patch [$F_{(2, 8)} = 34.61$, $p = 0.000115$]; (5) average

patch area [F(2, 8) = 6.019, *p* = 0.0254]; (6) sum of the naturalness [F(2, 8) = 02.383, *p* = 0.154] (Figure S1B).

Here we describe the observable characteristics for each cluster:

- Cluster 1 was composed by only one sub-window (A03) and identified as an outlier. The sub-window has a large patch of secondary vegetation in what would be a pasture dominated matrix. By implementing two SPS patches (of 4 and 5 ha), the PC index increased by 18%.
- Cluster 2 was composed of four sub-windows (A04, A07, A11, and A12) (Figure 4B). Their average forest habitat with the SPS implementations was 15%. The sub-windows grouped in this cluster are characterized by having a low percentage of crucial areas, but with different values of naturalness (secondary vegetation, fragmented forest with pastures and crops, fragmented forest with secondary vegetation, rivers, dense highland forest, and riparian forest). The average change in the landscape connectivity (PC index) within each sub-window increased by 3%, when comparing before and after the implementation of SPSs.
- Cluster 3 was composed of three sub-windows (A09, A01, and A10) (Figure 4B) with an average of 22% crucial area habitat without SPS implementations. The farmscape of this cluster are characterized by having a large crucial area patch, with an average size of 44.5 ha and made up of secondary vegetation. In each farmscape, a total of four SPS plots were adopted, each measuring about 4 ha. The change in the landscape connectivity (PC index) with the implementation of the SPS plots was an average increase of 6%.
- Cluster 4 was composed of four sub-windows (A 02, A05, A06, and A08) (Figure 4B). The amount of crucial area habitat without the SPS implementation was on average 10% per sub-window. This group contains the farmscapes with the least amount of crucial areas. The average change in the landscape connectivity (PC index) within the sub-windows with the implementation of SPSs was an increase of 5%.

The four clusters identified, showcase the variability that exists in the landscape in terms of habitat fragmentation and connectivity. Cluster 2 was characterized by an average isolation and an average habitat amount, where the addition of the SPS plots greatly contributed to the landscape connectivity. Cluster 3 was characterized by a low isolation and a high initial habitat amount, i.e., a landscape with a low fragmentation and well-connected remaining patches. In this case the SPS implementations did not have a relevant influence on the connectivity of the landscape, as the landscape was already well connected. In Cluster 4, the landscape was characterized by highly isolated patches and a low amount of habitat.

Finally, the change of connectivity (ΔPC) comparing the two conditions (with and without SPSs) varied between clusters. For example, in Clusters 2 and 4, characterized by the establishment of small implementation areas in the sub-window (SPS lots < 5 ha), few new patches (between one to three lots in the whole sub-window), and a distance between the SPS areas and the crucial areas of a maximum of 0.5 km, the increment in the dPCflux and dPCconnect was slightly lower than in cluster 3.

In contrast, Cluster 3, where the distance between patches does not exceed 300 m and where the number of large crucial habitat patches doubled (approx. 37.5 ha new areas in each sub-window, with an average SPS plot size of 12 ha), there was a greater increase in the DPCFlux. An increase in DPCFlux means that the flow between the patches improves. Basically, that there are more patches available to contribute to the dispersal of species. Likewise, an increase in the DPCconnector was observed (increases on average 72% in the sub-windows of this cluster), which indicates that these new patches are contributing as new connection areas (stepping stones) in the landscape. According to the results in the sub-windows A09, A01, and A10, the new silvopastoral implementations have the capacity to increase not only the flow but also the strength of the connections between the patches, helping facilitate the dispersal of species.

## 4. Discussion

During the last two decades (2001 to 2021), forest areas in the Colombian Andean-Amazon ecoregion have been reduced by more than 60%. This rate of deforestation is alarming as it disrupts the connectivity between two important biomes, the Andes mountain forests and the Amazon basin forests. From the results, it is clear that after the peace agreement was signed between the Colombian government and the Revolutionary Armed Forces (FARC) in 2016, deforestation rates increased reaching a historical high of 250,000 ha in 2018. This was likely a consequence of both, (i) the non-implementation of territorial management agreements in the peace agreement by the incoming government (period 2018–2022), and (ii) the illegal occupation of disenfranchised FARC dissidents (non-signatories of the peace agreements), lawless criminal bands and large-scale cattle ranchers with strong political connections, as well as newly engaged coca producers [39]. The observed socio-political restructuring and socio-ecological dynamics in the region have already been reported following the peace agreement signing [40,41].

Habitat loss in the Colombian Amazon ecoregion has been characterized by a reduction and fragmentation of natural areas such as the dense highland forests, the riparian and gallery forests, and the secondary forests and vegetation. As a consequence, landscapes dominated by pastures are evident today. Habitat destruction and fragmentation have driven many animal populations into remnant patches of varying size and isolation. Within this region, the mountain landscape (i.e., the Andes-Amazon corridor) was the most affected by habitat loss and fragmentation. In this landscape, the dense highland forests located in the transition area between the Alto Fragua Indiwasi protected area and the lomerio landscape, have been the most affected by increasing their patch isolation and reducing their patch size. These forests were reduced in terms of total area, patch area, increased perimeter, and porosity. Consequently, it increased the edge effect and reduced the total area available as habitat for wildlife. The combined impacts of habitat loss and fragmentation has severe consequences to wildlife and forest specialist species become more exposed and vulnerable to predation and hunting following habitat fragmentation [42].

Moreover, in the lomerio landscape, the loss of crucial areas has led to patches more isolated, smaller in size, with larger perimeter vs. core proportions, and less connected to each other. The lomerio landscape is dominated by highly degraded pastures, yet there are a significant number of secondary vegetation areas in the process of natural regeneration. Some of these secondary vegetation areas are protected by farmers involved in different conservation projects in the region. However, this commitment to conservation is always made voluntarily, and due to various circumstances, especially cultural and economic, farmers end up burning or cutting down these areas of vegetation for the implementation of pastures or crops [43,44]. By demonstrating the benefits of intensification practices in agricultural production and through the diversification of income sources and practices, little by little, or one tree at a time, the benefits of restoration and conservation activities may become more apparent to farmers, in order to reduce additional deforestation and to promote the regeneration of forest areas within productive landscapes.

It is clear that the introduction of arboreal areas in a degraded landscape can increase connectivity. Based on different authors [21,22,45], we know that the presence of living fences in pastures and agricultural areas can help reduce soil erosion, provide habitat to a variety of animal groups, and facilitate bird movement across the fragmented landscape, including several forest specialist species, such as the plain-brown woodcreeper (*Dendrocincla fuliginosa*). However, how other variables, such as the amount of habitat, patch size, distance between patches, or patch quality (its "naturalness" as per Machado's ranking [35]), can influence the probability of connectivity of a landscape. Our results provide evidence on how each of these variables influence the landscape connectivity. We found that small implementation areas with a large distance between the patches will lead to a smaller increase in the flow of the system, that is, the capacity for the patches to receive and disperse species. Furthermore, these characteristics will result in patches with less capacity to serve as stepping stones and connect crucial areas in the landscape. By contrast,

the implementation of larger patches in the landscape, which are also arranged at shorter distances from the crucial areas, will result in an increase in the flow of the system, and therefore, in a greater probability of connectivity of the landscape.

According to Calle, 2020 [14], the implementation of SPSs seems to be most favorable in landscapes with a high degree of fragmentation and a low habitat amount, i.e., landscapes such as the lomerio landscape in Caquetá, where the SPSs were implemented in our research. Our results would agree with this study. However, neither in Calle's investigation nor through this research, was an undegraded control area established. This is one of the limitations of the present study, which could be considered in future studies. In addition, it is important to be clear that this work is a geographical simulation of the potential of SPSs to increase landscape connectivity, but not an actual quantification of the connectivity increases, following the implementation of SPS systems in the degraded landscape. While silvopastoral systems based on the implementation of scattered trees in pasture plots and living fences is far from representing the restoration of a forest ecosystem, the PC index showed encouraging differences when comparing the two conditions simulated in GIS (with and without SPS implementations). This suggest that in cattle ranching landscapes, silvopastoral systems are a promising alternative to gradually introduce ecological restoration activities in regions where farmers have more than 40 years of conventional cattle ranching experience.

## 5. Conclusions

Land cover conversion from native vegetation to extensive and poorly managed pastures, negatively impacts the structural connectivity of the landscape, leading to ecosystem degradation. Combining cattle ranching and trees in SPSs has shown to be valuable in restoring highly fragmented landscapes. For farmers, these systems provide an opportunity to increase productivity and improve their well-being. This study examined the potential of SPSs to decrease habitat isolation and restore functional connectivity through the evaluation of the state of fragmentation and structural connectivity, before and after their adoption. This assessment was possible due to the availability and accessibility of the geographic data and robust free access software, such as FRAGSTATS, CONEFOR, and QGIS. While the true impact of SPSs on functional connectivity and biodiversity conservation can only be confirmed through field monitoring data, the present study provides spatially explicit insight on the effects of their implementation in improving the structural connectivity of the landscape.

Our findings suggest that the contributions of SPSs to landscape connectivity are not linear. There are other variables that must be considered, which could play a fundamental role when planning landscape restoration activities to enhance biodiversity conservation. According to the farmscape characteristics within the SPS implementation sub-windows, some of the main variables to estimate the potential impacts on connectivity include: the amount of initial habitat, the distribution of habitat patches, and the distance between habitat patches. In other words, the strategic adoption and implementation of SPSs could be co-designed to maximize pasture productivity for cattle ranching, while optimizing environmental benefits, such as biodiversity conservation.

In conclusion, connectivity and fragmentation assessments could be utilized in decision making and prior to the implementation of conservation actions, to increase benefits. The evaluation approach described here could contribute to evidence-based policy development, by providing information on priority sites to increase connectivity through the implementation of new areas of SPSs or agroforestry systems, that could ultimately lead to faster positive conservation impacts. It is key to engage local communities, and co-design with them conservation agreements to protect natural habitats within their properties and throughout the landscape. As a final recommendation, it is important to align landscape-based initiatives with other existing conservation programs in the region, both from NGOs and from the government, and in the Andes-Amazon ecoregion, those initiatives that will be developed within the framework of the peace agreement implementation.

**Supplementary Materials:** The following supporting information can be downloaded at: https://www.mdpi.com/article/10.3390/d14100846/s1, Table S1: Landscape unit's composition in the 2200 km$^2$ analysis window; Table S2: Landcover areas by landscape unit in the 2200 km$^2$ analysis window; Table S3: Area-edge, Subdivision, and Aggregation metric results of FRAGSTATS, by landscape for 2002; Table S4: Area-edge, Subdivision, and Aggregation metric results of FRAGSTATS, by landscape for 2018; Figure S1: (**A**) Optimal number of clusters. Average of Silhouette value vs. Number of cluster (k). (**B**) K-Means clustering algorithm and Silhouette score.

**Author Contributions:** Conceptualization, W.F. and K.A.; formal analysis, K.A. and B.R.-S.; investigation, K.A.; resources, K.A.; data curation, K.A.; writing—original draft preparation, K.A.; writing—review and editing, W.F.; visualization, K.A.; supervision, W.F. Project leader: M.Q.; research design K.A. and M.Q., fund acquisition M.Q. All authors have read and agreed to the published version of the manuscript.

**Funding:** This research contributes to the Sustainable Amazonian Landscapes Project (14_III_057_A_Latin America Sustainable Development Options). Grant number 42206-6157. These projects are part of the International Climate Initiative (IKI). The Federal Ministry for the Environment, Nature Conservation and Nuclear Safety (BMU) supports this initiative on the basis of a decision adopted by the German Bundestag. This study is framed in the CGIAR Research Program on Water Land Ecosystems (WLE).

**Institutional Review Board Statement:** Not applicable.

**Data Availability Statement:** Not applicable.

**Acknowledgments:** This research was undertaken as part of the project "Sustainable Amazonian Landscapes". This project is part of the International Climate Initiative (KI). The German Federal Ministry for the Environment, Nature Conservation, Building and Nuclear Safety (BMUB) supports this initiative based on a decision adopted by the German bundestag. CIAT, as administrator, leader and co-executor of the project, thanks the local partners (CIPAV-Center for Research on Sustainable Agricultural Production Systems, SINCHI-Amazonic Institute of Scientific Research and the University of the Amazon) for their great effort and support in the field to fully meet all of the goals established in the project. Finally, we express our gratitude to the local families from the municipalities of Albania, Belen de los Andaquies, and Morelia in Caquetá (Colombia) for their time and cooperation to land cover/land use map collection data, to the geographical delimitation of the areas of SPS implementations and to accomplish the project objectives.

**Conflicts of Interest:** The authors declare no conflict of interest.

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
