# Peer review of "One Tree at a Time: Restoring Landscape Connectivity through Silvopastoral Systems in Transformed Amazon Landscapes"

_diversity, doi:10.3390/d14100846_

Round 1

Reviewer 1 Report

see the uploaded file (in Word)

Both PDF files, they are intended to help the authors to improve the manuscript. I would expect the authors to answer the corrections and questions I made in the word file.

Author Response

Thank you very much for all your comments, they were really very valuable to improve our article. Hope it's much clearer now!

Author Response

(The authors gave the same response as above.)

Round 2

Reviewer 1 Report

The manuscript now is much easier to understand, and all methodological details are explained better. The addition of the new Figure 2 facilitates the explanation of the several methods used by the authors in this manuscript. The authors responded very well all my questions and suggestions and in its present form I recommend to publish this manuscript as it is in its new revised version. However, there are some minor changes that need to be done before publication; in the following lines I explain this minor changes one by one. It is my opinion that Figure 5 of the new version should not be included in the main body of the manuscript. I suggest to include this figure as a Supplementary Information figure since it shows methodological aspects of the cluster analysis and the results shown in the graph called “Cluster plot” (panel B of Figure 5) are not very clear; the clustering or grouping of this analysis is clearly shown in panel B of the next Figure (i.e. Figure 6B in the new version), and thus Fig. 5B and Fig.6 B include both the sites (i.e. sub-windows) grouped in the 4 resultant clusters.

The minor corrections detailed below are also included in the PDF file of the revised version of the manuscript. The PDF file with the new version has a little bit of disarray in the numbering of the lines (line numbers are not sequential throughout the whole file), and thus I provide the page number and the line number of that page, in which the correction has to be made.

Minor corrections:

Page1; line #57: delete the word “the” before landscape

Pg.3; ln #309 to 312: there are inconsistencies in the way in which “kilometer” is abbreviated, sometimes the authors write “Km” (with Capital K) and in other parts write “km” as it should be throughout the text. Please modify it, and be consistent. Also they wrote in line #309: “…a 41km x 54 km window…”, placing the unit’s “km” after each number, however in line #312 they wrote: “…3X3Km…” without repetition of the unit “km” after each number using capital K in the abbreviation. They should write instead: “…3 x 3 km…”

Pg.4; ln #387-390: write scientific names in italics (this has to be done for the whole manuscript).

Pg.7; ln #578: write “…These units shared…”; instead of “…Polygons that shared…”

Pg.8; ln #693: “…dominant…”; instead of “dominated”

Pg.12; ln #164: instead of “…by agriculture and pastures.”; write: “…by pastures and crop fields.” (or simply “…by agriculture…”). In English the word “agriculture” includes both cattle raising (or any type of livestock raising) and crop fields; thus, to say “dominated by agriculture and pastures” is incorrect because in English pastures are not a different type of activity than agriculture.

Pg.13; ln #287: write “…periods…” (in plural) instead of “period”.

Pg15; ln #632-633: Figure 5 should be moved from the main text to the Supplementary Information section. Both panels show mostly methodological aspects of the analysis and not results per-se, the cluster graph in panel B of this figure, shows the 4 clusters in an ordination plot (or similar to an ordination plot), but the resultant 4 clusters and the sub-windows grouped within each cluster are shown also in Fig. 6B. One aspect that I would like to remark is that the new version of this manuscript now explain more clearly what the authors did in this clustering. In particular the text written in lines #471 to 482 (Page 14), explains clearly what the authors did and find in this analysis without the need to show this part in a figure. Therefore, I still recommend (as I did in my first revision) to move this figure (both panels) into the Supplementary Information section. The main results of this analysis is clearly explained in the text and supported in Figure 6 panel B, and there is no need to add Fig. 5 to the main text.

Pg.16; ln #746: write “…the farmscapes of this cluster are characterized by…” (i.e. farmscapes in plural).

Pg.16; ln#753: write “…farmscapes…”; instead of “farmscape”.

Pg.16; ln #764: “…amount of habitat…”

Pg16; ln #770: write “…0.5 km…”, instead of “0.5Km”. Also the word “was” is written twice (delete one).

Pg.17; ln #987: write “…consequences of the demobilization…”

Pg.17; ln #1019: “…larger proportion of perimeter…”; instead of “larger in perimeter”

Pg.18; ln #1084: the correct common name of this bird is: “…the plain-brown woodcreeper…”

Pg.19; ln #1225: add the word “of”, it should say: “…the implementation of new areas…”

See attached PDF file (second revision) with my comments highlighted in yellow (each correction is marked in this PDF file).

Author Response

Thank you very much for all your suggestions both in the first round of major changes and in this second round. The changes greatly contributed to improving the structure, flow and comprehension of the article.

This is my first article, and now seeing this version I feel satisfied with the result. I really appreciate your input.

I am attaching the new clean version for easy reading.

Thanks again,

Karolina Argote

Reviewer 2 Report

Thank you for a very thorough review that has made the paper much easier to follow.  It now seems quite clear and I have no outstanding comments.  

Author Response

Thank you very much for your review, your suggestions and comments. They contributed to improve this article.
I am attaching a clean version to facilitate the final reading of the document
Cordially,
Karolina Argote
